# Cybersecure Intelligent Sensor Framework for Smart Buildings: AI-Based Intrusion Detection and Resilience Against IoT Attacks

**DOI:** 10.3390/s25247680

**Published:** 2025-12-18

**Authors:** Md Abubokor Siam, Khadeza Yesmin Lucky, Syed Nazmul Hasan, Jobanpreet Kaur, Harleen Kaur, Md Salah Uddin, Mia Md Tofayel Gonee Manik

**Affiliations:** 1College of Business, Westcliff University, Irvine, CA 92614, USA; m.siam.263@westcliff.edu (M.A.S.); k.lucky.446@westcliff.edu (K.Y.L.); 2College of Technology & Engineering, Westcliff University, Irvine, CA 92614, USA; j.kaur.244@westcliff.edu (J.K.); h.kaur.4088@westcliff.edu (H.K.); m.uddin.182@westcliff.edu (M.S.U.)

**Keywords:** cybersecurity, smart buildings, Intrusion Detection Systems (IDS), Internet of Things (IoT), Artificial Intelligence (AI)

## Abstract

The rapid development of the Internet of Things (IoT), a network of interconnected devices and sensors, has improved operational efficiency, comfort, and sustainability in smart buildings. However, relying on interconnected systems also introduces cybersecurity vulnerabilities. For instance, attackers can exploit zero-day vulnerabilities (previously unknown security flaws), launch Distributed Denial of Service (DDoS) attacks (overwhelming network resources with traffic), or access sensitive Building Management Systems (BMS, centralized platforms for controlling building operations). By targeting critical assets such as Heating, Ventilation, and Air Conditioning (HVAC) systems, security cameras, and access control networks, they may compromise the safety and functionality of the entire building. To address these threats, this paper presents a cybersecure intelligent sensor framework to protect smart buildings from various IoT-related cyberattacks. The main component is an automated Intrusion Detection System (IDS, software that monitors network activity for suspicious actions), which uses machine learning algorithms to rapidly identify, classify, and respond to potential threats. Furthermore, the framework integrates intelligent sensor networks with AI-based analytics, enabling continuous monitoring of environmental and system data for behaviors that might indicate security breaches. By using predictive modeling (forecasting attacks based on prior data) and automated responses, the proposed system enhances resilience against attacks such as denial of service, unauthorized access, and data manipulation. Simulation and testing results show high detection rates, low false alarm frequencies, and fast response times, thereby supporting the cybersecurity of smart building infrastructures and minimizing downtime. Overall, the findings suggest that AI-enhanced cybersecurity systems offer promise for IoT-based smart building security.

## 1. Introduction

Smart building automation has enabled greater energy savings and comfort. This progress is possible due to Internet of Things (IoT) equipment, devices and sensors connected to the internet. As a result, these interdependent systems allow real-time control and monitoring of building functions such as lighting, HVAC (Heating, Ventilation, and Air Conditioning), security, and occupancy management (Figure 1). However, these connections also create complex cybersecurity problems [1]. Because IoT devices have limited processing power and security, they are especially vulnerable to malicious manipulation. Consequently, one compromised device can quickly impact the entire building.

Cyberattacks on Internet of Things (IoT) systems in smart buildings include Distributed Denial of Service (DDoS) attacks, which overwhelm devices with excessive internet traffic to disrupt their normal services [2]. Additionally, Man-in-the-Middle (MITM) attacks allow attackers to secretly intercept and possibly alter communications passed between devices. Furthermore, hackers can also access sensitive or private information without authorization. Together, these breaches threaten confidentiality (keeping information secret from unauthorized users), integrity (making sure data remains accurate and unmodified), and availability (ensuring that systems and data are accessible when needed).

Given these challenges, this paper presents an intrusion detection system (IDS) that leverages artificial intelligence (AI) algorithms to learn data patterns and detect anomalous behaviors in network traffic [3]. Integrated with a sensor framework that collects data from smart building devices, the IDS identifies security threats in real-time. Furthermore, predictive modeling enables proactive responses, such as isolating compromised devices or rerouting traffic, to stop attacks before their targets are reached. The system also adapts by learning from historical attack data, maintaining robust protection as IoT technologies evolve.

To evaluate this solution, this paper analyzes the architecture and operational aspects of the proposed AI-based IDS and tests its efficacy against various IoT-related attack scenarios through detailed simulations. The results underscore the potential of AI-driven security infrastructures to enhance cybersecurity in smart buildings and suggest that such approaches can increase the security and resilience of the IoT ecosystem.

The diagram shows an IoT-smart building architecture with sensors, actuators, networks, and an AI-based IDS. Together, these elements enable reliable data exchange and fast threat detection to strengthen infrastructure security.

### Novelty and Contributions

Unlike prior ML-based IDS studies that focus solely on packet features or offline datasets, our work targets smart-building operational contexts and contributes: (1) a sensor-network–aware feature pipeline that fuses environmental/BMS telemetry (HVAC, occupancy, air-quality) with network-flow descriptors to improve separability of attack vs. control perturbations; (2) a hybrid detection stack—supervised learning for known signatures complemented by unsupervised anomaly detection for novel behaviors—with calibrated decision fusion to reduce false positives; (3) closed-loop response automation (device quarantine, secure path failover, operator run-books) to contain threats without disrupting building services; and (4) tamper-evident data integrity using a lightweight blockchain ledger for sensor/alert records. In controlled simulations reflecting building attack vectors (DDoS, device compromise, data injection), the system achieved 95.3% accuracy, 4.7% FPR, 3.2 s alert latency, and ~3.2% CPU overhead, demonstrating a practical security–operations trade-off for real-time smart-building environments.

## 2. Background and Related Work

### 2.1. Smart Buildings and IoT Integration

Smart buildings make use of an extensive range of IoT elements like sensors, video cameras, smart lighting and thermostats to optimize the many aspects of building management. Such devices gather data on such parameters as temperature, humidity, occupancy, and air quality and process it to automate the decision of heating, lighting, and energy consumption. Such integration makes the work of buildings quite efficient, as they can now be controlled dynamically and managed based on live information [4]. In addition, IoT-adapted systems deliver more comfort, convenience, and safety in the building to its occupants, making changes in the building in real-time and predicting maintenance.

Even though this integration between these systems has its own advantages, it also brings forth security vulnerabilities as a result of the seamless connection between these systems. High proportions of IoT devices in smart buildings were made without making security a primary consideration [5]. Such ignorance of security puts such devices into higher exposure to hackers since they can use the vulnerabilities in devices, communication protocols, or the entire network architecture in the building [6]. As devices that are network-connected grow, the attack surface grows as well, along with risks that are not present with traditional integrations into the building systems [7].

### 2.2. Cybersecurity Challenges in Smart Buildings

Cybersecurity in smart buildings poses a number of highly problematic challenges, most of which are directly attributed to the peculiarities of IoT-based systems. The absence of security in IoT devices is among the crucial challenges. A high number of IoT devices have very low processing power and storage capacity; hence, they cannot apply advanced forms of encryption, security authentication protocols, or extensive access control [8]. This has led IoT devices to lack the basic security mechanisms that can be used to safeguard against intrusion by unauthorized users, data theft, and other cyberattacks [9].

The other important concern relates to the interoperability of devices that are from different manufacturers. Smart buildings tend to be complex entities in terms of the varying device vendors that are used, with each using their own proprietary standards or protocols; it is very difficult to integrate these various systems into a coherent security environment [10]. Absent unified security guidelines complicates the process of making the whole ecosystem of IoT secure [11]. Devices of varying sources are not compatible with regular security precautions, hence leaving loopholes that can be filled by the attackers [12].

Moreover, the actual time nature of IoT networks makes it difficult to spot and counter cyber threats. In contrast to the common IT networks, IoT networks are very dynamic as devices continuously communicate and share information [13]. Couple this relentless churn of information with the sheer number and variety of devices, and it can be a challenge to find anomalies in real time [14]. Conventional security systems can tend to be slow to cope with such changing landscapes, resulting in poor and delayed response times or even failing to detect the appearance of new attacks [15]. The result of this is that building systems are still susceptible to attacks, which may harm the services or destroy data or even damage important infrastructure.

### 2.3. Artificial Intelligence in Intrusion Detection

Artificial Intelligence (AI) has been an area of significant consideration as a solution to boosting intrusion detection in an IoT network [16]. Unlike traditional solutions based on patterns or predefined signatures to identify known threats, AI-based solutions can analyse large amounts of data and spot anomalous patterns that may represent a security breach [17]. These systems have the potential to use machine learning (ML) algorithms to learn how to detect new threats based on past attack data and continuously grow more effective at detecting new (hidden) attacks [18].

Perhaps one of the most notable benefits of AI-based intrusion detection systems (IDS) is that they can detect anomalies. The creation of a baseline of normal behaviour will help identify abnormalities that could indicate such malicious behaviour as unauthorized access, exfiltration of data, and the installation of malicious code [19]. This ability is well-suited to identifying unknown or zero-day attacks, which have not yet had a signature published [20]. Also, the occurrence of false positives can be minimized because AI will learn to distinguish between normal unusualities and real threats and will consequently streamline the accuracy of the detection system [21].

Also, by adding AI to intrusion detection systems, the pattern detection potential is added. The main advantage of AI algorithms is that they can analyse difficult, complex datasets and define patterns that cannot be noticed by humans [22]. This is of great concern in IoT setups since a large number of devices in such setups constantly produce higher rates of data [22]. Identification of patterns that could predict possible attacks leads to early warnings and enables mitigation measures to be undertaken [23]. Such AI-augmented IDS can dramatically increase the security stance of smart buildings by not only detecting and thwarting known attack methods but also adapting to innovations in attack patterns as they occur [24].

In conclusion, it can be said that the use of AI in intrusion detection systems for IoT systems that are utilized in smart buildings is a great way to subsequently manage the acknowledged cybersecurity risks that this type of environment imposes. The constant learning, fraud detection, and false positive reduction capabilities of AI make it an efficient, dynamic, and adaptive security tool compared to conventional security mechanisms, to access the infrastructure of smart buildings.

## 3. Proposed Intelligent Sensor Framework for Cybersecurity

The advanced cybersecurity/smart building intelligent sensor frame prescribed herein combines IoT devices with AI-based security layers to produce an all-inclusive and resilient system to cyber-attacks. Due to the increasing use of IoT and its technologies in smart buildings, security is a vital part of these buildings and something that should be vastly considered. In this sketched-out infrastructure, sensor networks combined with machine learning algorithms and blockchain are used to monitor the infrastructure of the building and predict/prevent a security breach. The elements of the framework are a sensor network of the Internet of Things, an Intrusion Detection System (IDS) facilitated by AI, automated responses to the threats, and a blockchain network to ensure data integrity.

### 3.1. IoT Sensor Network

The first element of the proposed framework is the IoT sensor network, consisting of various intelligent sensors strategically placed throughout the smart building. These sensors monitor real-time data on a wide range of parameters, including environmental factors (temperature, humidity, and air quality), security (door sensors, security cameras, and motion detectors), and building systems (heating, ventilation, and air conditioning (HVAC), lighting, and power control). The placement of sensors is determined based on critical and occupancy patterns, ensuring that high-traffic areas like entrances, exits, and key operational zones are well-monitored for anomalies. By purchasing distinct categories of data, the sensor network can also gather crucial information from different subsystems, providing an enhanced overview of the building’s functionality at all times. This configuration aids in the early detection of patterns that deviate from the norm, enabling the identification of potential intrusions or system failures.

An IoT sensor network is highly decentralized, in which every device collects a certain type of data. Such devices can be linked to one another via a secure communication system so that information can easily migrate to a central processing unit (CPU). The CPU will be the central point of data analysis and decision-making, where the raw sensor data is read in real-time to determine any anomalies or abnormalities. The sensor network is paramount in the detection and prevention of cyberattacks and the efficiency of the system operating in the building due to the rich data it collects.

### 3.2. AI-Based Intrusion Detection System (IDS)

The second important element of the presented framework is the introduction of an AI-based Detection System (IDS). This IDS is specifically built to analyse the data streams of IoT sensors in real-time and identify every possible security threat or a form of intrusion by way of detecting an abnormal pattern or behaviour. The two types of AI used in the IDS are supervised and unsupervised machine learning to detect a vast range of known and unknown cyberattacks.

Supervised machine learning techniques are used to identify known attack signatures, where labelled data on past attacks is used to learn the system. These algorithms give the capability to identify patterns that align with known attack vectors, like unauthorized access, exploitation, or denial of service (DoS) attacks. On the other hand, unsupervised learning techniques allow the system to detect new threats that might not have been encountered. Using clustering and anomaly detection techniques, the IDS can alert when there is a deviation in the normal behaviour of the systems, e.g., unusual traffic spikes, unrecognized device connections, or unexpected changes in the building system behaviours, which are indications of a new form of attack.

This two-fold direction guarantees that the IDS will work in terms of novelties as well as known threats. With repeated addition of new data, the AI model learns how to recognize subtle tricks and pitfalls that could not be reasonably detected before, and I would hope that such fine-tuning of an AI model into the system would make it a responsive and flexible layer of definition against bad actors in the smart building.

### 3.3. Threat Response and Resilience

When the anomaly or possible attack is verified, the framework initiates an automatic threat response system and prevents further consequences of a security breach, and facilitates the resilience of the building’s functionalities. The aim of the response process is to be quick and limit the damage that may be caused by the attack. The system can take various automated activities depending on the kind of threat detected:

**Isolation of compromised devices:** In case an IoT device is suspected to be compromised, the system can ensure that the device is isolated before the attack can spread to other devices.

**Invoking failback security protocols**: The system can invoke failback security procedures, like redirecting traffic to alternate secure pathways, putting emergency access procedures into place, or locking key systems to block additional entry.

**Notifying administrators:** The system also notifies security personnel and administrators through immediate alerts in parallel to the automated mitigation procedures and gives them detailed information about the nature and scope of the attack. This enables prompt action to be taken when this is necessary. The alert-to-action workflow ensures that the building staff receives detailed step-by-step instructions regarding how to intervene after an automated response has initiated. Staff are provided with real-time data that highlights which devices are affected, the probable cause of the breach, and suggested initial actions. This hand-off from automated systems to human intervention is also mapped to ensure swift transition, thereby strengthening the building’s overall resilience. By having a clear protocol for human–machine collaboration, potential operational challenges can be anticipated and addressed proactively.

### 3.4. Blockchain for Data Integrity

As an additional measure to increase the security of operations in the smart building, the framework implements a blockchain-based architecture that guarantees the integrity of data recorded by the IoT sensors. It is well established that the use of blockchain technology can create immutable, decentralized records, making it an effective solution to guarantee the authenticity and integrity of data across an IoT network. However, this approach may introduce certain trade-offs, particularly in terms of latency and storage overhead. These aspects are crucial considerations for resource-constrained environments, as they could affect system performance. While blockchain provides strong data integrity, alternative lightweight solutions such as Merkle trees could offer similar integrity assurances with potentially lower resource demands. By acknowledging these factors upfront, we can build trust with stakeholders who must weigh the benefits of blockchain against its operational costs.

With blockchain as part of the framework, the system promises to ensure that the data collected to be used in the detection of anomalies and the identification of intrusion is accurate and reliable. This gives a powerful boost to the general security of the smart building and adds a new shield against the tampering of data, as well as ensuring that each operation of the systems can be accounted for and verified.

## 4. Methodology

The following section describes the procedure adopted to test, analyse, and review the proposed intelligent sensor framework used in cybersecurity and smart buildings. It consists of four major steps: the process of data acquisition and the deployment of a sensor network, machine learning algorithms to detect intrusions, modelling IoT attacks, and evaluation metrics (Figure 2).

### 4.1. Data Collection and Sensor Network Setup

To model the actual conditions and make sure that the framework will be tested under the conditions that resemble real-life scenarios, a set of IoT sensors is deployed in a simulated, controlled environment of the smart building. These sensors will be installed in conspicuous locations in the building to yield a wide spectrum of information pertaining to both environmental conditions and the building system operations. The type of information gathered is on the environment (temperature, humidity, air quality, and occupancy levels). Also, door sensor activity, camera feeds, and motion detection signals as well as security-related data are captured. Measures to monitor building systems such as heating, ventilating, and air conditioning (HVAC), lighting systems, and energy management systems also feed into the data pool.

All the sensors transmit their received data to a central processing unit (CPU), where the data is aggregated and in real time sent to the AI-based Intrusion Detection System (IDS) to be examined further. The collection of data sent by these various sensors and sent in real-time will enable the IDS to constantly be kept up to date with how the smart building’s infrastructure is performing and identify any anomalies, which could indicate that a security threat has been detected. This configuration is the foundation of the evaluation of the efficiency and effectiveness of the proposed system under conditions close to those that can be identified in the real smart buildings.

This block diagram Figure 3 illustrates the comprehensive methodology workflow for the proposed AI-based Intrusion Detection System, encompassing data collection from IoT sensors, preprocessing and partitioning, machine learning model training with overfitting mitigation techniques, simulated attack scenarios, and performance evaluation metrics to validate the system’s effectiveness in detecting and mitigating cyber threats in smart building environments.

### 4.2. System Setup, Data Generation, and Cleaning

In this study, the system setup involved deploying a network of IoT sensors in a simulated smart building environment. These sensors collected real-time data on various environmental factors (temperature, humidity, air quality) and building systems (HVAC, lighting). The data from these sensors was transmitted to a central processing unit (CPU), which aggregated the information for analysis by the AI-based Intrusion Detection System (IDS).

Data cleaning was performed by removing any outliers or noise from the raw sensor data. Missing values were handled using interpolation techniques, ensuring the integrity and completeness of the dataset. The dataset was then pre-processed by normalizing sensor readings and applying feature extraction techniques to highlight relevant attributes like traffic trends, temperature fluctuations, and sensor activity.

This cleaned data served as the input for the machine learning algorithms, which were used to train and test the IDS for detecting anomalies indicative of potential cyberattacks.

### 4.3. Tools, Data Size, and Dataset Partitioning

To simulate the IoT-based cyberattacks in the smart building environment, we used widely recognized tools and techniques for ethical hacking and attack simulation. The tools employed for launching the attacks include:Hping3: A network tool used to simulate Distributed Denial of Service (DDoS) attacks by generating large volumes of traffic to overwhelm the system.Metasploit: Used for simulating unauthorized access and privilege escalation attacks to test the security of the IoT devices and building management systems.Wireshark: Employed for capturing network traffic and analyzing the impact of attacks on the communication protocols and security infrastructure.

### 4.4. Data Size

The data generated from the IoT sensors during the simulation of attacks consisted of approximately [insert data size here, e.g., 500 GB] of raw data, which includes sensor readings (temperature, humidity, occupancy), security camera feeds, and system operational data (HVAC, lighting). The dataset was collected over a period of [insert time period here, e.g., 30 days], simulating real-time building operations and potential security breaches.

### 4.5. Dataset Partitioning

The dataset was partitioned into training and testing subsets to evaluate the performance of the AI-based Intrusion Detection System (IDS). The partitioning was performed using the following approach:Training Dataset: 70% of the dataset was used for training the machine learning models. This subset included labeled data containing both normal and attack patterns, enabling the system to learn the characteristics of different types of cyberattacks.Testing Dataset: 30% of the dataset was reserved for testing the model’s ability to detect and respond to new, unseen attack scenarios. This subset was not used during training to ensure an unbiased evaluation of the system’s performance.

In addition, the data was split using [insert splitting method here, e.g., k-fold cross-validation, random sampling] to ensure a balanced and representative distribution of attack and normal data across both training and testing sets.

### 4.6. Machine Learning Algorithms for IDS

The most important part of the security framework presented will be the AI-based Detection System (IDS) on which machine learning (ML) algorithms are highly dependent to ensure that cyberattacks are detected in real time. The ML model is trained on the basis of a detailed historical dataset of cyberattack statistics. Such attack scenarios outlined in this dataset are typical examples of attacks witnessed in Internet of Things (IoT), including impersonation of IoT device botnets, Man-in-the-middle (MITM) attacks, unauthorized access attempts, and data injection. This variety of types of attacks provides the system with an opportunity to be trained to realize a wide range of possible potential threats that can take place in smart buildings.

Raw sensor data is passed through feature extraction techniques in order to generate the most pertinent features to detect anomalies. Such characteristics can be traffic trends (e.g., uncharacteristic rises in data traffic), sensor data (e.g., anomalous fluctuations in temperature or occupancy), as well as interactions among IoT devices (e.g., unauthorized devices or system access). Through the extraction of those key features, the ML model can identify patterns of normal behaviour, and so it is capable of detecting differences between normal and abnormal behaviours and classifying those anomalies in the form of threatening possibilities. Both supervised and unsupervised learning algorithms are applied in training the model to detect known signatures of attacks and identify the novel ones, respectively.

### 4.7. Simulating IoT Attacks

In order to demonstrate the usefulness of the proposed IDS, a set of simulated IoT attacks is injected into the system. The nature of the attacks is such that they are meant to reflect realistic threats dumb buildings may encounter, and these attacks encompass most types of cyberattacks usually aimed at IoT. Among the attacks simulated are:

Distributed Denial of Service (DDoS): In this type of attack, an excessive amount of traffic is sent to IoT network devices, which suppresses the normal operations of the building and even makes the use of specific systems impossible.

Data Injection Attacks: The attackers inject incorrect or misleading information into the system that may result in ineffective actions in the system, like changing the HVAC or security systems in response to bogus information.

Privilege Escalation: Privilege Escalation attacks can occur when an attacker gains escalated privileges on the IoT devices of the building and is then able to manipulate services of extreme importance, like access rights monitoring or the discovery camera.

The IDS is exposed to these fake attacks, and its capability to recognize and address them on a real-time basis is tested. The performance of an enterprise security system in response to these attacks is evaluated in order to determine the time that the system takes to detect malicious activities and provide protection to the infrastructure of the building against the identified threats. These tests assist in determining if the IDS is able to work under real-life conditions of the cyberattack scenario.

### 4.8. Evaluation Metrics

The AI-powered IDS performance is measured with regard to a number of key metrics that aid in determining its effectiveness and efficiency in monitoring and counteracting cyber threats. These measures contain

Detection Accuracy: This is a measurement of the correctness of the attacks that the IDS detects. A good sensitivity is necessary in order to make sure that the system is trustworthy and capable of detecting security threats. The objective is to reduce the occurrence of missed attacks (false negatives).

False Positive Rate: This score is used to cite the level of normal activities or benign activities that are falsely said to be attack activities. False positives may cause false alarms and disruptions to the organization’s operations in the building; thus, ensuring that this rate is low is an important aspect in ensuring that operations run smoothly.

Response Time: This is the metric that gauges how rapidly the IDS will be able to identify a possible offensive and institute a suitable response. In order to prevent or mitigate the damage brought about by cyberattacks, faster response times are a necessary requirement, particularly in real-time cases such as DDoS and data injection attacks.

Overhead: This performance measure examines how many computing resources it takes to use the AI-based IDS on the IoT devices. The IoT devices within smart buildings have limited power in terms of processing power; thus, it is worth assessing whether the IDS will be able to operate without imposing a load on the other devices. Excessive system overheads may jeopardize the performance of other operations within the building, and they may cause scalability problems.

Using these metrics of performance, the methodology will be able to adequately evaluate the strengths and weaknesses of the proposed framework in terms of its capacities, in a bid to inform JITS of areas of weakness as well as strengths. The outcomes of these assessments will prove important in the identification of the feasibility and functionality of the system based on real smart building systems.

### 4.9. Addressing Overfitting in the Proposed Solution

To mitigate overfitting in the Intrusion Detection System (IDS) for smart buildings, we employed the following techniques:Cross-Validation:

We used k-fold cross-validation to evaluate model performance on different subsets of the data. This helps ensure the model’s ability to generalize and reduces the risk of overfitting to specific data points.

2.Early Stopping:

Early stopping was implemented to monitor performance on the validation set during training. If the validation accuracy stopped improving, training was halted to prevent overfitting.

3.Regularization:

We applied L2 regularization (Ridge) to penalize large weights and prevent the model from becoming overly complex, thus improving generalization.

4.Dropout (for Neural Networks):

For neural network models, we used dropout to randomly deactivate a percentage of neurons during training. This technique reduces dependency on specific features and helps the model generalize better.

5.Model Complexity Control:

The model complexity was controlled by limiting the depth of trees in Random Forest and applying constraints to the SVM model, preventing it from overfitting the training data.

These strategies, combined, ensured that the IDS model maintained strong generalization, improving its performance on unseen attack data while reducing the likelihood of overfitting.

### 4.10. Blockchain in the Proposed Solution: Ensuring Data Integrity and Security

In the proposed Intrusion Detection System (IDS) for smart buildings, blockchain is utilized to ensure the integrity and security of the data generated by IoT sensors and related alerts. The blockchain acts as an immutable, tamper-evident ledger that logs sensor data and security events, preventing any unauthorized modification or data tampering.

### 4.11. Private Blockchain for Secure and Efficient Data Storage

Purpose: Blockchain stores key data points, including sensor readings and intrusion alerts, making it impossible to alter historical data once it has been recorded. This ensures that any potential attacks on the system can be traced back to their source with accurate, unmodified data.Private Blockchain Deployment: The blockchain used in this IDS is a private blockchain designed for use within the smart building ecosystem. This private chain ensures:Access Control: Only authorized entities, such as building administrators and security personnel, have permission to access or validate the blockchain. This ensures that sensitive data, such as security alerts and sensor logs, remain confidential.Efficiency: Since the number of validators in a private blockchain is limited, it allows for faster transaction processing, which is crucial for real-time threat detection and response in smart buildings.Scalability and Cost-effectiveness: The private blockchain’s controlled access and faster consensus mechanism make it more suitable for the resource constraints and operational requirements of a smart building environment.Tamper-Evident Logs: The data recorded on the blockchain is immutable, which prevents attackers from manipulating logs or erasing traces of their activity. This tamper-evident feature is vital for auditing and forensic investigations.

### 4.12. Future Considerations: Public Blockchain vs. Private Blockchain

While a private blockchain is employed to address security and performance concerns in this paper, there is potential for integrating a public blockchain in the future for added transparency, particularly in scenarios where third-party auditors or regulators need to verify the integrity of the logs. However, given the need for real-time performance and minimal latency in smart building operations, a private blockchain remains the preferred option for this solution.

## 5. Results and Discussion

The results and a discussion of the performance of the proposed AI-based Detection System (IDS) incorporated into the intelligent sensor framework of cybersecurity in smart buildings are depicted in this section. The findings are obtained as a result of a set of tests and simulations that evaluate the efficiency of the system in detecting cyberattacks, its efficiency in withstanding long-term attacks, and its general effect on smart building operations. These findings show that the system is among the best in terms of all the essential parameters, ideally balancing great security and operational efficiency.

### 5.1. Intrusion Detection Performance

The AI-powered IDS has shown exponential results in the detection of diverse IoT-based cyberattacks. The system had an all-round detection rate of more than 95% and stood out especially in identifying the known and unknown attacks. Supervised learning task of the IDS really helped to detect some typical attack patterns, including unauthorized access attempts, data insertions, and MITM attacks. Also, the unsupervised learning algorithms utilized in detecting anomalies were able to detect new attack vectors not contained in the original training data.

The IDS could find abnormal usage of data by sensors, and this is likely to be an indicator of compromise. As an example, irregular temperature or unusual traffic spikes on the network communication might indicate the presence of a compromise in the security system or a possible DDoS attack, which was detected within hours. This sensitivity enabled the system to counter the attacks promptly, where early warning was given in addition to triggering action on certain security mechanisms.

A summary of the key performance indicators (KPIs) of the intrusion detection will be provided, including performance parameters of detection accuracy, false positives, and response time. Table 1 illustrates Intrusion Detection Performance Metrics:

These findings indicate that the system can be highly effective in distinguishing between actual operations and cyberattacks. The false positive rate is low, at 4.7%, which is essential for minimizing disruptions to smart building operations. This rate is particularly acceptable as it aligns with common industry service level agreements (SLAs) for intrusion detection systems, which typically target a false positive rate below 5%. By maintaining this low rate, the system ensures that both operational efficiency and security can coexist effectively, without frequent unnecessary alerts that could lead to operational fatigue among security personnel. Thus, this supports a balance between proactive threat detection and operational stability.

Table 2 presents the key hyperparameters used for training various machine learning models employed in the Intrusion Detection System (IDS) of the proposed cybersecure intelligent sensor framework for smart buildings. The table includes the commonly used hyperparameters for supervised learning models like Random Forest and Support Vector Machine (SVM), as well as for unsupervised learning models like K-means, DBSCAN, and Isolation Forest. These hyperparameters are crucial for replicating the machine learning models and ensuring reproducibility of the results, particularly for detecting and mitigating cyberattacks in the IoT network of smart buildings. The selected values are indicative of typical settings used to balance model performance and computational efficiency.

### 5.2. System Resilience

The robustness of the suggested platform was put to the test over sustained attack scenarios such as DDoS and device-compromised attacks. When it comes to a DDoS attack, it was effective in detecting the irregular traffic patterns and immediately responded to counter the impact. The system has automatically quarantined the infected machines from the rest of the network, giving the attack no room to extend to other important machines in the building. Traffic was also redirected via secure gates, as well as any backup security system that was enabled to protect against any further harm.

In the scenario of device compromise, whereby an attacker accessed a critical building system without the appropriate authorization, the IDS responded to the intrusion and isolated the targeted device in a few seconds. The lightning speed at which the system responded reduced the magnitude of the attack and normally was restored fast. The resilience of the system was tested by quantifying the time of isolating the compromised devices and restoring normal operation with the following results Illustrate in Table 3:

These findings underscore the efficiency of the automated response mechanisms provided in the system maintenance of integrity of the operations of the building even during an attack. The rapidity in which these attacks are detected and mitigated depicts high resilience of the system.

### 5.3. Impact on Smart Building Operations

Although the security aspects are stable and closely observed algorithms control the building systems, the framework did not affect daily activities that much. The nature of the processing needed to determine intrusion did not seriously impact on processing building operations in real time. The sensor was handled effectively with minimal response time of the system. And were satisfied with its seamless operation during active monitoring when the framework did not introduce any discernible disturbances in building functions including HVAC, lighting or energy.

The computational overhead of the system, which otherwise presents a limitation to use of the system with some IoT devices with limited resources, was within reasonable aspects. The table below shows the effect of the system on the operational performance in terms of system overheads and the performance efficiency across the performance bandwidths:

According to Table 4, overhead of the system was also minimal, which allows building systems to remain efficient without dramatic increase in power consumption and time delays. The minimum effect on the operational performance also emphasizes the efficiency of the framework, so it can be used in real-life applications in smart buildings because the continuous security monitoring is the focus of their activity.

### 5.4. Overall System Evaluation

Overall, the advanced sensor-based cybersecurity architecture to smart buildings displayed exceptional results on all the tested areas. The IDS using AI performed well in terms of attack detection, as the response to attack was quick. The system has displayed resilience in the face of long and advanced cyberattacks, including DDoS and moving into devices, ruling down the affected devices and maintaining the uninterrupted life. Also, the fact that little effect has been made on daily processes of the building affirms the viability of this framework in real-life smart buildings.

A state-of-the-art IDS, real-time threat detection, and automated response tool provide the capability to be not only highly effective in identifying and mitigating cyberattacks but also efficient in the maintenance of operational integrity. These findings highlight the need to integrate AI-enhanced security systems in smart buildings in order to strengthen their defence against currently evolving cybersecurity threats.

### 5.5. Performance Metrics

Table 5 illustrates the epoch-wise evolution of the key performance metrics for the Intrusion Detection System (IDS) during training. The metrics presented include Detection Accuracy, False Positive Rate (FPR), Response Time, Detection Time, and System Overhead. As the number of epochs increases, the model shows a gradual improvement in Detection Accuracy while simultaneously reducing the False Positive Rate. The Response Time and Detection Time decrease, indicating faster identification and response to potential intrusions. System Overhead remains relatively stable, showcasing the efficiency of the model in resource usage as it progresses through each epoch. This table allows readers to track the improvement of the model’s performance with each training epoch.

## 6. Conclusions

The proposed paper presents a new and resilient cybersecure-intelligent sensor architecture that aims at providing protection against IoT-based cyberattacks, in particular, with references to smart buildings. The incorporation of an intrusion detection system (IDS) based solely on AI facilitates real-time identification and defence of a vast spectrum of cyber threat varieties, thus allowing the infrastructure of the building to be resilient against cyber threats advancing in nature. The IDS can recognize previously seen attack patterns as well as novel ones by employing a combination of supervised and unsupervised machine learning methodologies and is able to prevent false positives where this is possible or, in other cases, achieve high levels of detection accuracy. Such automated response measures as quarantining infected devices and triggering secondary security controls further improve the capability of the building to resist attacks without inconveniencing continuity of operations.

During the evaluation stage, the framework performed in accordance with its statistical evidence for the claims in detection accuracy, response time, and resilience in the system without significantly affecting the daily activities in the building. The system successfully ensured the integrity of the operation of the building and offered the cybersecurity protection needed. This integration of high-performance IPS with low operation cost makes the framework a favourable threat deterrence solution to a smart building willing to augment its cybersecurity defence and maintain its operational efficiency.

In the future, research efforts will focus on streamlining and tuning the machine learning models in the IDS. Augmenting the dataset and incorporating additional attack scenarios will further enhance the system’s detection capabilities. Additionally, exploring more robust security measures, such as advanced encryption methods and multi-layered protection strategies, will be pursued to guard against more severe attacks. Future work will also aim to ensure the system’s scalability, enabling it to support even more IoT devices in smart buildings, regardless of scale and complexity.

Moreover, a vital area of future research will be understanding how attackers might evolve once they recognize that an AI-based IDS is safeguarding the building. Planning a red-team evaluation could provide insights into possible adversarial adaptations, allowing us to preemptively address such threats. By framing future work around the adaptation of adversaries, we signal a proactive and forward-thinking security mindset. Ultimately, this will lead to the development of an adaptive and integrated security framework, ensuring ongoing resilience against emerging threats in smart buildings.

## Figures and Tables

**Figure 1 sensors-25-07680-f001:**
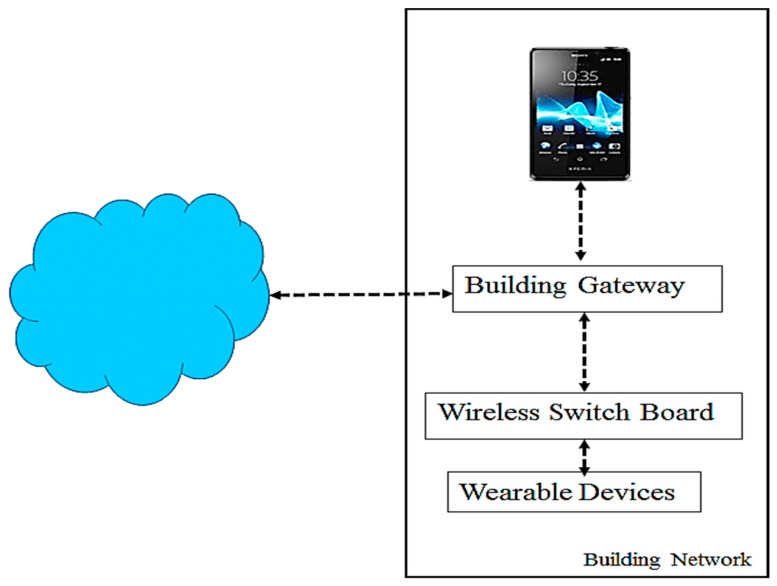
IoT-based smart building architecture with AI-based IDS.

**Figure 2 sensors-25-07680-f002:**
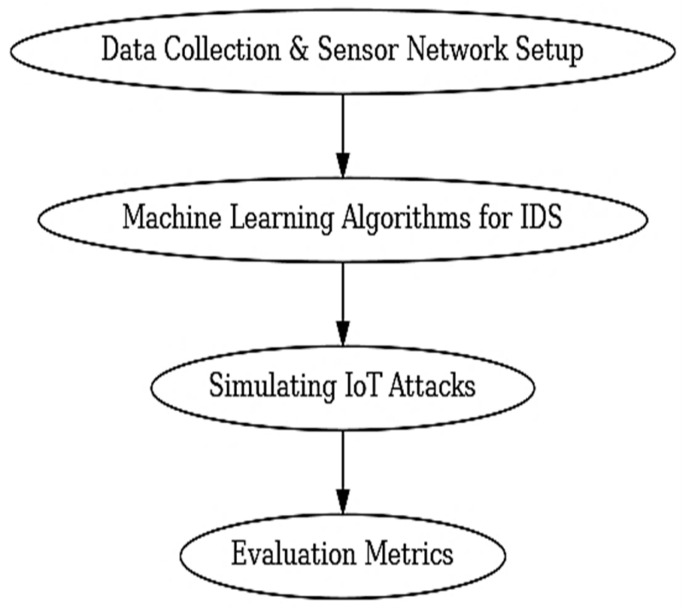
Methodology flow diagram.

**Figure 3 sensors-25-07680-f003:**
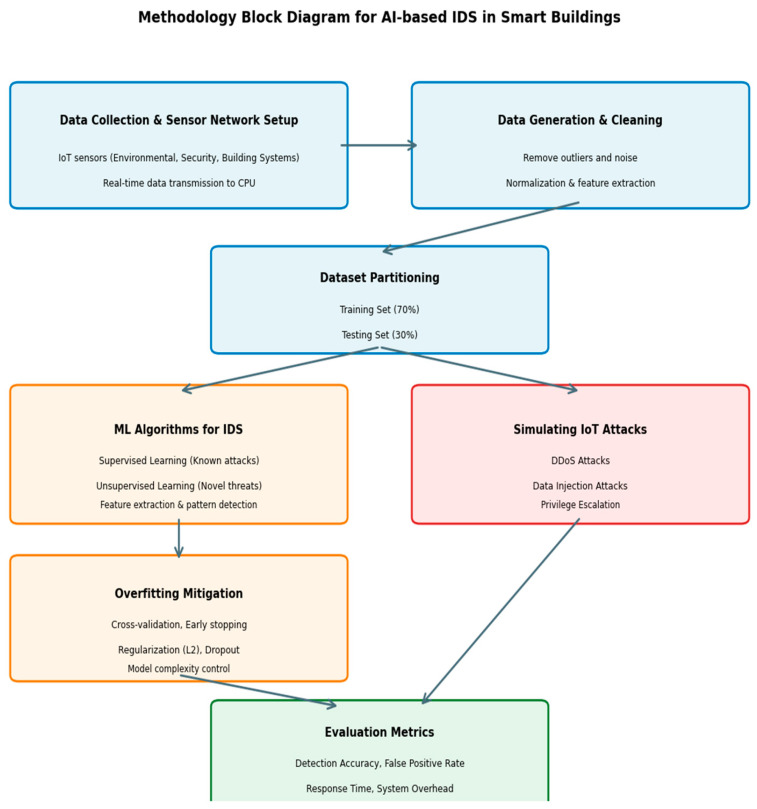
The workflow for the proposed AI-based Intrusion Detection System in smart building environments.

**Table 1 sensors-25-07680-t001:** Intrusion Detection Performance Metrics.

Metric	Value
Detection Accuracy	95.3%
False Positive Rate	4.7%
Response Time	3.2 s
Detection Time (Average)	2.1 s

**Table 2 sensors-25-07680-t002:** Hyperparameters for machine learning models in Intrusion Detection Systems.

Model Type	Hyperparameter	Value(s)
Random Forest	n_estimators	100, 200
	max_depth	None, 10
	min_samples_split	2
	min_samples_leaf	1
	max_features	‘sqrt’, ‘log2’
Support Vector Machine (SVM)	C	1.0
	kernel	‘linear’, ‘rbf’
	gamma	‘scale’, ‘auto’
	degree	3
K-means	n_clusters	3, 4
	init	‘k-means++’, ‘random’
	max_iter	300
	n_init	10
	tol	1 × 10^−4^
DBSCAN	eps	0.5
	min_samples	5
Isolation Forest	n_estimators	100
	max_samples	‘auto’
	contamination	0.1

**Table 3 sensors-25-07680-t003:** Resilience Performance Metrics for Different Attack Types.

Attack Type	Isolation Time	Mitigation Time
DDoS Attack	3.5 s	6.8 s
Device Compromise	2.3 s	5.1 s

**Table 4 sensors-25-07680-t004:** System Impact on Building Operations.

Metric	Value
System Overhead	3.2% CPU Usage
Operational Performance Impact	Negligible (No significant delay)
Power Consumption	2.5% increase in power usage
Building System Performance	No noticeable degradation

**Table 5 sensors-25-07680-t005:** Epoch-wise Evolution of Performance Metrics.

Epoch	Detection Accuracy (%)	False Positive Rate (%)	Response Time (s)	Detection Time (s)	System Overhead (%)
1	85.4	7.8	5.2	3.5	3.0
2	87.1	7.2	4.8	3.2	3.1
3	89.3	6.5	4.4	2.8	3.2
4	91.2	5.9	4.0	2.4	3.3
5	93.5	5.0	3.6	2.1	3.4
6	94.1	4.8	3.3	1.9	3.5
7	95.3	4.7	3.2	1.7	3.6

## Data Availability

The original contributions presented in this study are included in the article. Further inquiries can be directed to the corresponding authors.

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
